# Zero Shot Recommender Systems

**Hao Ding**[1]**, Anoop Deoras**[1]**, Yuyang Wang**[1]**, Hao Wang**[2][1]
[1]AWS AI Labs, [2]Rutgers University
{haodin, adeoras, yuyawang}@amazon.com, hw488@cs.rutgers.edu

## Abstract

Performance of recommender systems (RecSys) relies heavily on the amount of training data available. This poses a chicken-and-egg problem for early-stage products, whose amount of data, in turn, relies on the performance of their RecSys. In this paper, we explore the possibility of zero-shot learning in RecSys, to enable generalization from an old dataset to an entirely new dataset. We develop, to the best of our knowledge, the first deep generative model, dubbed ZEro-Shot Recommenders (ZESRec), that is trained on an old dataset and generalize to a new one where there are *neither overlapping users nor overlapping items*, a setting that contrasts typical cross-domain RecSys that has either overlapping users or items. We study three pairs of real-world datasets and demonstrate that ZESREC can successfully enable such zero-shot recommendations, opening up new opportunities for resolving the chicken-and-egg problem for data-scarce startups or early-stage products.

## 1 Introduction

As machine learning models, the performance of RecSys relies heavily on the amount of training data available. This might be feasible for large e-commerce or content delivery websites such as Overstock and Netflix, but poses a serious chicken-and-egg problem for small startups, whose amount of data, in turn, relies on the performance of their RecSys. On the other hand, zero-shot learning Xu et al. (2020); Snell et al. (2017) promises a certain degree of generalization from an old dataset to an entirely new dataset. In this paper, we investigate the topic of zero-shot learning in RecSys and start by formally identifying *four key properties* of this problem: *(1) cold users, (2) cold items, (3) domain gap, and (4) no access to target data* (see section 2 for details).

In this paper, we propose, to the best of our knowledge, the first deep generative model, dubbed ZEro-Shot Recommenders (ZESREC), to address this problem. Combining the merits of sequential RecSys (solve cold users), the idea of universal continuous ID space (solve cold items), and a novel deep generative model (mitigate domain gap), our ZESREC successfully enables recommendation in the zero-shot setting where all users and items in the target domain are unseen during training. Essentially ZESREC tries to learn transferable user behavioral patterns in a universal continuous ID space. To summarize our contributions:

- We identify the problem of zero-shot recommender systems and introduce the notion of universal continuous identifiers that makes recommendation in a zero-shot setting possible.
- We propose ZESREC as the first deep generative model for addressing this problem, and derive two Bayesian inference schemes based on our vanilla ZESREC.
- We provide empirical results, demonstrating the effectiveness of both the vanilla and the two Bayesian versions of ZESREC in the zero-shot recommendation setting.
- We provide case studies showcasing that ZESREC can learn interpretable user behavioral patterns that can generalize across datasets.

## 2 Zero-Shot Recommender Systems

In this section we introduce our ZESREC which is compatible with any sequential model. Without loss of generality, here we focus on NL descriptions as a possible instantiation of universal identifiers, but note that our method is generalizable to other modalities such as images and videos.

**Definition of Zero-shot Learning in RecSys.** Our zero-shot setting includes four unique properties: **(1) Cold Users:** No overlapping users between the training data and the test data. **(2) Cold Items:** No overlapping items between the training data and the test data. **(3) Domain Gap:** The training and test data come from different domains (i.e., a source domain and a target domain). **(4) No Access to Target Data:** Target-domain data is available only during inference, and it only allows online access.

**Problem Setup.** A model is trained using user-item interactions from the source domain and then deployed for personalized recommendation in the target domain given user's history. Note that in practice we append a dummy item at the beginning of each user session, so during inference we could recommend items even for users without any history by ingesting the dummy item as context. In our zero-shot setting, the model is *not allowed to fine-tune on any data from the target domain*.

## 2.1 FROM CATEGORICAL DOMAIN-SPECIFIC ITEM ID TO CONTINUOUS UNIVERSAL ITEM ID

Current RecSys models learn item embeddings through interactions. These embeddings are indexed by *categorical domain-specific item ID*, which is transductive and not generalizable to unseen items. In this paper, we propose to use item generic content information such as NL descriptions to produce item embeddings, which can be used as *continuous universal item ID*. Since such content information is domain agnostic, the model trained on top of it would be transferable across domains, therefore making zero-shot RecSys feasible. Based on universal item embeddings, one can then build sequential models to obtain user embeddings by aggregating embeddings of items in user histories. Therefore we introduce universal embedding networks (UEN), which use continuous universal embeddings to index items (item UEN) and users (user UEN); these will be the backbones of our generative models.

## 2.2 MODEL

We propose a deep generative model with a probabilistic encoder-decoder architecture. The encoder ingests items from user history to yield the user embedding, while decoder computes recommendation scores based on similarity between user embeddings and item embeddings.

**Generative Process.** The generative process of ZESREC (in the source domain) is as follows (see figure 1 for the corresponding graphical model):

1. For each item $j$:
   - Compute the item universal embedding: $\mathbf{m}_j = f_e(\mathbf{x}_j)$.
   - Draw a latent item offset vector $\boldsymbol{\epsilon}_j \sim \mathcal{N}\left(\mathbf{0}, \lambda_v^{-1}\mathbf{I}_D\right)$.
   - Obtain the item latent vector: $\mathbf{v}_j = \boldsymbol{\epsilon}_j + \mathbf{m}_j$.
2. For each user $i$:
   - For each time step $t$:
     - Compute the user universal embedding: $\mathbf{n}_{it} = f_{seq}([\mathbf{v}_{i_\tau}]_{\tau=1}^{t-1})$.
     - Draw the latent user offset $\boldsymbol{\xi}_{it} \sim \mathcal{N}\left(\mathbf{0}, \lambda_u^{-1}\mathbf{I}_D\right)$.
     - Obtain the latent user vector: $\mathbf{u}_{it} = \boldsymbol{\xi}_{it} + \mathbf{n}_{it}$.
     - Compute recommendation score $\mathbf{S}_{itj}$ for each tuple $(i, k, j)$, $\mathbf{S}_{itj} = f_{softmax}(\mathbf{u}_{it}^\top\mathbf{v}_j)$ and draw the $t$-th item for user $i$: $\mathbf{R}_{it*} \sim Cat([\mathbf{S}_{itj}]_{j=1}^J)$.

Here $f_{softmax}(\cdot)$ is the softmax function: $f_{softmax}(\mathbf{u}_{it}^\top\mathbf{v}_j) = \exp(\mathbf{u}_{it}^\top\mathbf{v}_j)/\sum_j \exp(\mathbf{u}_{it}^\top\mathbf{v}_j)$. $Cat(\cdot)$ is a categorical distribution. $f_e(\cdot)$ is item UEN, $f_{seq}(\cdot)$ is user UEN. $i_\tau$ in $\mathbf{v}_{i_\tau}$ indexes the $\tau$-th item that user $i$ interacts with. $\lambda_u$ and $\lambda_v$ are hyperparameters. The latent item offset $\boldsymbol{\epsilon}_j = \mathbf{v}_j - \mathbf{m}_j$ provides the final latent item vector $\mathbf{v}_j$ with the flexibility to slightly deviate from the content-based item universal embedding $\mathbf{m}_j$. Similarly, the latent user offset $\boldsymbol{\xi}_{it} = \mathbf{u}_{it} - \mathbf{n}_{it}$ provides the final latent user vector $\mathbf{u}_{it}$ with the flexibility to slightly deviate from the user universal embedding $\mathbf{n}_{it}$. Intuitively, $\boldsymbol{\epsilon}_j$ and $\boldsymbol{\xi}_{it}$ provide domain-specific information on top of the domain-agnostic information from $\mathbf{m}_j$ and

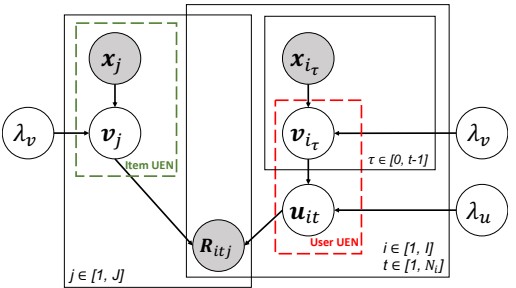

Figure 1: Graphical model for ZESREC. The item side (left) and the user side (right) share the same $\lambda_v$ and $\mathbf{v}$'s. The plates indicate replication.

$\mathbf{n}_{it}$. In the target domain we will remove $\boldsymbol{\epsilon}_j$ from $\mathbf{v}_j$, and $\boldsymbol{\xi}_{it}$ from $\mathbf{u}_{it}$, which can be seen as an attempt to remove the bias learned from the source domain.

**Training.** The MAP estimation in the source domain can be decomposed as following:

$$P(\mathbf{U}^{(s)}, \mathbf{V}^{(s)}|\mathbf{R}^{(s)}, \mathbf{X}^{(s)}, \lambda_u^{-1}, \lambda_v^{-1}) \propto P(\mathbf{R}^{(s)}|\mathbf{U}^{(s)}, \mathbf{V}^{(s)}) \cdot P(\mathbf{U}^{(s)}|\mathbf{V}^{(s)}, \lambda_u^{-1}) \cdot P(\mathbf{V}^{(s)}|\mathbf{X}^{(s)}, \lambda_v^{-1}),$$

where $\mathbf{U}^{(s)}, \mathbf{V}^{(s)}, \mathbf{R}^{(s)}$ and $\mathbf{X}^{(s)}$ denote the collection of all users, items, user-item interactions, and NL descriptions of items in the *source domain*, respectively.

Maximizing the posterior probability is equivalent to minimizing the joint Negative Log-Likelihood (NLL) of $\mathbf{U}^{(s)}$ and $\mathbf{V}^{(s)}$ given $\mathbf{R}^{(s)}, \mathbf{X}^{(s)}, \lambda_u^{-1}$, and $\lambda_v^{-1}$:

$$\mathcal{L} = \sum_{i=1}^{I_s}\sum_{t=1}^{N_i} -\log(f_{softmax}(\mathbf{u}_{it}^\top \mathbf{v}_{it})) + \frac{\lambda_u}{2}\sum_{i=1}^{I_s}\sum_{t=1}^{N_i} ||\mathbf{u}_{it} - f_{seq}(\{\mathbf{v}_{i_\tau}\}_{\tau=1}^{t-1})||_2^2 + \frac{\lambda_v}{2}\sum_{i=1}^{J_s} ||\mathbf{v}_j - f_e(\mathbf{x}_j)||_2^2. \quad (1)$$

Training on the source-domain data produces MAP solutions with corresponding UENs:

$$(\mathbf{U}_{MAP}, \mathbf{V}_{MAP}) = \operatorname*{argmax}_{\mathbf{U}, \mathbf{V}} p(\mathbf{U}, \mathbf{V}|\mathbf{X}) \approx \Big(f_{seq}(f_e(\mathbf{X})), f_e(\mathbf{X})\Big). \quad (2)$$

**Inference and Recommendation in the Target Domain.** Once the model is trained using source-domain data, it can recommend unseen items $j \in \mathcal{V}_t$ (where $\mathcal{V}_t \cap \mathcal{V}_s = \emptyset$) for any unseen user $i \in \mathcal{U}_t$ (where $\mathcal{U}_t \cap \mathcal{U}_s = \emptyset$) from the target domain based on the approximate MAP inference below:

$$p(\mathbf{R}^{(t)}|\mathbf{X}^{(t)}) = \int p(\mathbf{R}^{(t)}|\mathbf{U}^{(t)}, \mathbf{V}^{(t)}, \mathbf{X}^{(t)})p(\mathbf{U}^{(t)}, \mathbf{V}^{(t)}|\mathbf{X}^{(t)})d\mathbf{U}^{(t)}d\mathbf{V}^{(t)}$$

$$\approx \int p(\mathbf{R}^{(t)}|\mathbf{U}^{(t)}, \mathbf{V}^{(t)}, \mathbf{X}^{(t)})\delta_{\mathbf{U}_{MAP}^{(t)}}(\mathbf{U}^{(t)})\delta_{\mathbf{V}_{MAP}^{(t)}}(\mathbf{V}^{(t)})d\mathbf{U}^{(t)}d\mathbf{V}^{(t)},$$

where $\delta(\cdot)$ denotes a Dirac delta distribution. $\mathbf{U}_{MAP}^{(t)}$ and $\mathbf{V}_{MAP}^{(t)}$ are the MAP estimate of $\mathbf{U}^{(t)}$ and $\mathbf{V}^{(t)}$ given $\mathbf{X}^{(t)}$ in the *target domain*, using the learned functions $f_{seq}(\cdot)$ and $f_e(\cdot)$. The reason for the approximation is that ZESREC has no access to interactions $\mathbf{R}^{(t)}$ in the target domain, making the posterior collapse to the prior. The user and item latent matrices $\mathbf{U}_{MAP}^{(t)}, \mathbf{V}_{MAP}^{(t)}$ in the target domain enable us to perform zero-shot recommendation by computing recommendation scores based on inner products and recommend item $\operatorname{argmax}_j f_{softmax}(\mathbf{u}_{it}^\top \mathbf{v}_j)$.

**Full Bayesian Treatment.** Besides MAP inference, we have derive a full Bayesian treatment for ZESREC. Experiments show that our full Bayesian version could further improve the performance of zero-shot recommendation (see section A.6 for details).

**Item UEN and User UEN.** The generative process above relies on the item UEN $f_e(\cdot)$ to obtain the item universal embedding $\mathbf{m}_j$ and the user UEN $f_{seq}(\cdot)$ to obtain the user universal embedding $\mathbf{n}_{it}$. In practice, these UENs instantiated using a pretrained BERT network coupled with a sequential model, e.g., an RNN. See section A.8 for implementation details.

## 3 EXPERIMENTS

In this section, we evaluate our ZESREC against various in-domain and zero-shot baselines on three source-target dataset pairs, with the major goals of addressing the following questions:

**Q1** How accurate (effective) is ZESREC compared to the baselines? (section 3.2)

**Q2** If one allows training models using target-domain data, how long does it take for non zero-shot models to outperform zero-shot recommenders? (section A.3)

**Q3** Does ZESREC yield meaningful recommendations for users with similar behavioral patterns in the source domain and target domain? (section A.4)

**Simulated Online Scenarios.** In our experiments, ZESRec accesses target-domain data only during inference (*but not during training*) to simulate online scenarios, where new businesses just open and the customers are using service in real-time. This online access setting is substantially different from batch access ahead of time as it prevents us from training a RecSys in target domains before serving.

**Datasets.** We use three different real-world dataset pairs with item NL descriptions, one from Amazon McAuley et al. (2015) and two from MIND (Wu et al., 2020). **(1) Amazon**: We consider

Table 1: Zero-shot results on three dataset pairs: (1) Amazon: 'Grocery and Gourmet Food' → 'Prime Pantry', and (2) MIND: 'News' → 'Finance', and 'Lifestyle' → 'Finance'. Methods such as HRNN, HRNN-Meta, and POP are *oracle methods* trained on target-domain data. The top 3 zero-shot results are shown in bold. N@20 represents NDCG@20 and R@20 represents Recall@20.

| Method | AMAZON G → P | | MIND N → F | | MIND L → F | |
|---|---|---|---|---|---|---|
| | N@20 | R@20 | N@20 | R@20 | N@20 | R@20 |
| HRNN (ORACLE) | 0.038 | 0.073 | 0.063 | 0.136 | 0.063 | 0.136 |
| HRNN-META (ORACLE) | 0.045 | 0.089 | 0.046 | 0.117 | 0.046 | 0.117 |
| GRU4REC (ORACLE) | 0.042 | 0.081 | 0.060 | 0.135 | 0.060 | 0.135 |
| GRU4REC-META (ORACLE) | 0.044 | 0.088 | 0.042 | 0.111 | 0.042 | 0.111 |
| TCN (ORACLE) | 0.038 | 0.073 | 0.061 | 0.136 | 0.061 | 0.136 |
| TCN-META (ORACLE) | 0.045 | 0.088 | 0.048 | 0.120 | 0.048 | 0.120 |
| POP (ORACLE) | 0.007 | 0.018 | 0.004 | 0.013 | 0.004 | 0.013 |
| EMB-KNN (BASELINE) | 0.024 | 0.042 | 0.022 | 0.057 | 0.022 | 0.057 |
| RANDOM (BASELINE) | 0.001 | 0.002 | 0.008 | 0.019 | 0.008 | 0.019 |
| ZESREC-H (OURS) | **0.027** | **0.051** | **0.029** | **0.079** | **0.025** | **0.075** |
| ZESREC-G (OURS) | **0.027** | **0.050** | **0.027** | **0.080** | **0.037** | **0.122** |
| ZESREC-T (OURS) | **0.028** | **0.052** | **0.028** | **0.073** | **0.025** | **0.076** |

pair 'Grocery and Gourmet Food' → 'Prime Pantry'. **(2) MIND**: In our experiments, we use the 1-week interaction data to simulate zero-shot learning by transferring knowledge from one category to another in MIND. We consider two pairs: (1) 'News' → 'Finance' and (2) 'Lifestyle' → 'Finance'.

**Experiment Setup.** We adopted a rigorous experiment setup to ensure (1) **no overlapping users and items** and (2) **no temporal leakage**. We performed temporal train-test split in the ratio of 8:2 for the target domain and prevent temporal leakage from the source domain. See section A.1 for more details about the data and section A.2 for more details about the data preprocessing).

**Evaluation Protocol.** For evaluation, we adopted Recall (R@20) and the ranking metric Normalized Discounted Cumulative Gain (NDCG) Shani & Gunawardana (2011) (N@20). We removed all the repetitive interactions (e.g., user A clicked item B two times in a row) to only focus on evaluating the model's capability of capturing the transition between user history to the next item.

## 3.1 BASELINES AND ZESREC VARIANTS

We compare ZESREC against two groups of baselines: in-domain methods and zero-shot methods.

**In-Domain Methods.** We compare variants of our model ZESREC against a variety of state-of-the-art session-based recommendation models including **GRU4Rec** Hidasi et al. (2015), **TCN** Bai et al. (2018), and **HRNN** Ma et al. (2020). We also consider their extensions, **HRNN-Meta**, **GRU4Rec-Meta**, and **TCN-Meta**, which use items' NL description embeddings to replace item ID hidden embeddings. We also introduce **POP** which recommends based on item popularity. All the above 7 methods are trained directly on target-domain data and therefore are considered *'oracle'* methods.

**Zero-Shot Methods.** Since no previous work has been done on this thread, we consider two intuitive zero-shot models (1) **EmbeddingKNN**: a K-nearest-neighbors algorithm based on the inner product between the user embedding (average of embeddings of interacted items) and item embedding (BERT embedding from text), and (2) **Random**: random item selection without replacement.

**ZESREC Variants.** We evaluate three variants of our ZESREC, including **ZESREC-G**, **ZESREC-T**, and **ZESREC-H** which use GRU4Rec, TCN and HRNN as base models, respectively.

## 3.2 ZERO-SHOT EXPERIMENT AND RESULTS

**Zero-Shot Experiments.** We trained in-domain baselines on target domain training set, while our ZESREC is trained on source domain. All models are tested on the testing set of the target domain for an apples to apples comparison.

**Zero-Shot Experimental Results** Table 1 shows the NDCG and Recall of different methods on three dataset pairs. Overall, our ZESREC outperforms zero-shot baselines Embedding-KNN and Random by a large margin in most cases; it can also achieve performance comparable to in-domain baselines.

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

## A APPENDIX

### A.1 DATASETS

**Amazon** McAuley et al. (2015): A publicly available dataset collection which contains a group of datasets in different categories with abundant item metadata such as item description, product images, etc. In our experiments, we consider two datasets: (1) 'Prime Pantry', which contains 300K interactions, 10K items, and 76K users, and (2) 'Grocery and Gourmet Food', which contains 2.3M interactions, 213K items, and 739K users.

**MIND** (Wu et al., 2020): A large-scale news recommendation dataset collected from the user click logs of Microsoft News. It includes 1-week user-item interactions and provides 4-week user history as context. In our experiments, we use the 1-week interaction data to simulate zero-shot learning by transferring knowledge from one category to another in MIND. We consider two pairs: (1) 'News' → 'Finance' and (2) 'Lifestyle' → 'Finance'. 'News' contains 434K interactions, 4.3K items, and 133K users; 'Lifestyle' contains 155K interactions, 0.8K items, and 55K users; 'Finance' contains 175K interactions, 1.1K items, and 61K users.

## A.2 DATA PREPROCESSING

We adopted a rigorous experimental setup for zero-shot learning to ensure (1) **no overlapping users and items** and (2) **no temporal leakage**. Specifically, we ensure that there are **no** overlapping users and items between the source domain and the target domain; meanwhile we temporally split the two domains such that all training interactions in the source domain must happen before all testing interactions in the target domain.

Datasets in all pairs are split using a time stamp threshold $t_s$. For target domain, we choose $t_s$ such that the data is divided into training (80%) and test (20%) sets. In order to prevent temporal leakage, we use the threshold $t_s$ from the target domain to split source-domain data. For all datasets, We further split 10% of the training sets by user as validation sets.

## A.3 INCREMENTAL TRAINING EXPERIMENTS

To measure how long it takes for non-zero-shot models to outperform zero-shot recommenders, we conducted incremental training experiments on in-domain base models GRU4Rec, TCN, HRNN as well as GRU4Rec-Meta, TCN-Meta, HRNN-Meta. Note that the variants of our ZESREC are NOT retrained on target domain. It is also *inevitable* that non-zero-shot models eventually outperform ZESREC because ZESREC does not have access to target-domain data.

For all the source-target dataset pairs, we group the interactions by user and sort interactions within each user based on interaction timestamps. We randomly select users until we get enough interactions and build three datasets containing 2.5K, 5K, and 10K interactions, respectively.

Note that the two MIND pairs have the same target domain 'Finance'; therefore we plot their incremental training results for the same metric in one figure (see figure 2 (a) and (b)).

**Incremental Training Experimental Results.** We plot the incremental training results in figure 2. We combined the results of two MIND pairs into figure 2 (a) and (b) since they have the same target domain 'Finance', where 'L ZESREC-H' and 'L ZESREC-T' represent ZESREC variants trained on the source domain 'Lifestyle' while 'N ZESREC-H' and 'N ZESREC-T' represent ZESREC variants trained on the source domain 'News'. The results of the Amazon pair are shown in figure 2 (c) and (d). Overall, all the in-domain baselines are unable to outperform ZESREC by retraining on at most 10K interactions in the target domain; the gap between retrained in-domain baselines and ZESREC is prominent on Amazon Prime Pantry, showcasing the critical importance of conducting zero-shot learning in RecSys. For new business operating an early-stage RecSys, it's hard to train a good RecSys with limited interactions. This is a chicken-and-egg problem, as training good RecSys requires sufficient interactions, while in turn, collecting sufficient interactions requires a satisfactory RecSys to attract users. Therefore the first 10K interactions are crucial to get the RecSys started.

## A.4 CASE STUDIES

**Experiment Procedure** To gain more insight what ZESREC learns, we perform several case studies. Specifically, we randomly select users from the test set of the target domain Amazon Prime Pantry (where we evaluate ZESREC) and only keep users for whom ZESREC correctly predicts the 6-th items in the sequence given the first 5 items as context, as we want to focus on sequences where ZESREC works. We use these users as queries to find users with similar behavioral patterns from the source domain Amazon Grocery and Gourmet Food (where we train ZESREC) based on user embeddings from ZESREC. User embeddings are generated based on the first 5 items of the sequence.

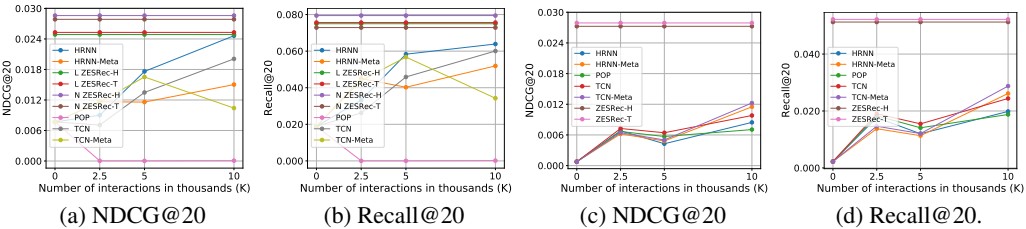

|     |     |     |     |
| --- | --- | --- | --- |
| (a) NDCG@20 | (b) Recall@20 | (c) NDCG@20 | (d) Recall@20. |

Figure 2: Incremental training results for baselines using target domain data compared to ZESREC using *no training data* on 'Finance' (left two) and 'Prime Pantry' (right two). To prevent clutter, we only show results for TCN-based and HRNN-based ZESREC, since HRNN has a similar architecture with GRU4Rec. Results show that even without using target-domain data, ZESREC can still outperform models trained directly using target-domain data for substantial amount of time.

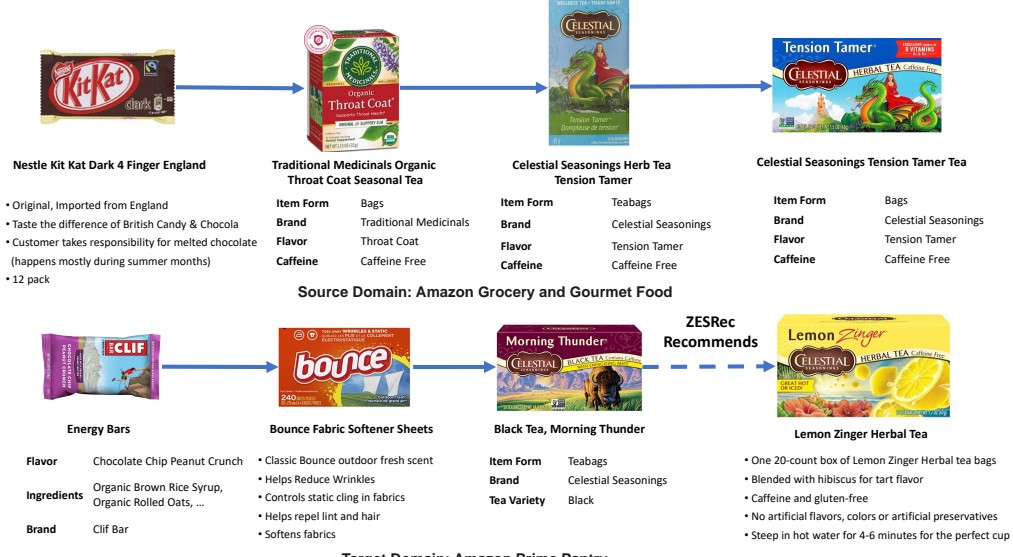

Figure 3: Case Study 1. The purchase history of a user in the source domain (*top*) and the purchase history of an unseen user in the target domain, where all items are unseen during training (*bottom*). We select two users with similar universal embeddings according to section A.4. This case study demonstrates ZESREC can learn the user behavioral pattern that 'users who bought sugary snacks and tea tend to buy caffeine-free herbal tea later'.

The goal of our case studies is to demonstrate our ZSR could learn relevant dynamics of users' purchase history from the source domain and successfully recommend unseen products to an unseen user in the target domain.

figure 3 shows the purchase history of a user in the source domain (*top*), 'Amazon Grocery and Gourmet Food', and the purchase history of an unseen user in the target domain (*bottom*), 'Amazon Prime Pantry', where all items are unseen during training. The user in the source domain bought 'Tension Tamer Tea', which is a type of herbal tea, after buying some sugary snacks (KitKat) and other tea. Such a pattern is captured by ZESREC, which then recommended 'Lemon Zinger Herbal Tea' to an unseen user after she bought some sugary snacks ('Energy Bars from Clif Bar') and some black tea. This case study demonstrates ZESREC can learn the user behavioral pattern that 'users who bought sugary snacks and tea tend to buy caffeine-free herbal tea later'. More interestingly, another case study in figure 4 demonstrates that ZESREC can learn the user behavioral pattern that 'if users bought snacks or drinks that they like, they may later purchase similar snacks or drinks with different flavors'. Specifically, in the source domain, the user purchased 'Vita Coconut Water' with four different flavors; such a pattern is captured by ZESREC. Later in the target domain, an unseen user purchase 'V8 Splash' with a tropical flavor, ZESREC then successfully recommends 'V8 Splash' with a berry flavor to the user.

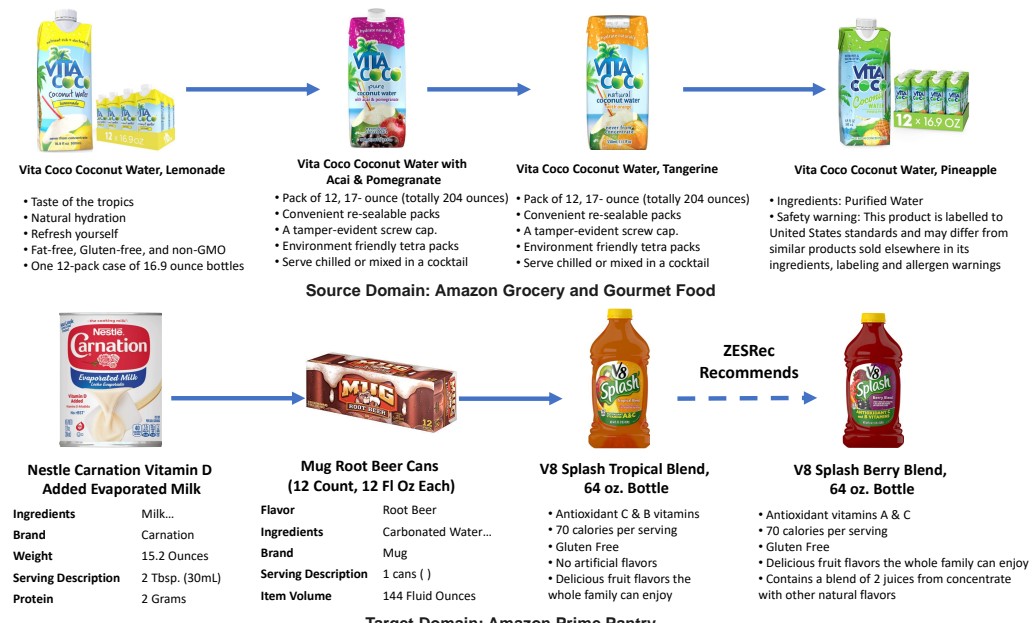

Figure 4: Case Study 2. The purchase history of a user in the source domain (*top*) and the purchase history of an unseen user in the target domain, where all items are unseen during training (*bottom*). We select two users with similar universal embeddings according to section A.4. This case study demonstrates ZESREC can learn the user behavioral pattern that 'if users bought snacks or drinks that they like, they may later purchase similar snacks or drinks with different flavors'.

## A.5 ABLATION STUDIES ON OUR BAYESIAN MODEL

To verify the domain debiasing effect of our Bayesian framework, we performed ablation studies on all three dataset pairs. In the experiments, we implemented a vanilla version of ZESREC where item offset vectors and user offset vectors are removed; in this case the item embedding will collapse to the prior text embedding. Table 2 shows the results for the vanilla ZESREC and the Bayesian ZESREC. On almost all cases the Bayesian ZESREC outperforms the vanilla ZESREC by a large margin, verifying the effectiveness of our Bayesian ZESREC.

Table 2: Ablation studies on the Bayesian framework. 'G → P' represents 'Grocery and Gourmet Food' → 'Prime Pantry', 'N → F' represents 'News' → 'Finance', and 'L → F' represents 'Lifestyle' → 'Finance'. N@20 and R@20 represent NDCG@20 and Recall@20, respectively. The best results for each ZESREC variant (ZESREC-H/G/T) are shown in bold.

| Method | AMAZON G → P | | MIND N → F | | MIND L → F | |
|---|---|---|---|---|---|---|
| | N@20 | R@20 | N@20 | R@20 | N@20 | R@20 |
| ZESREC-H (BAYESIAN) | **0.027** | 0.051 | **0.029** | 0.079 | 0.025 | 0.075 |
| ZESREC-G (BAYESIAN) | **0.027** | **0.050** | 0.027 | 0.080 | 0.037 | 0.122 |
| ZESREC-T (BAYESIAN) | **0.028** | **0.052** | 0.028 | 0.073 | 0.025 | 0.076 |
| ZESREC-H (VANILLA) | 0.027 | **0.052** | 0.025 | 0.062 | 0.014 | 0.033 |
| ZESREC-G (VANILLA) | 0.026 | 0.050 | 0.022 | 0.049 | 0.011 | 0.027 |
| ZESREC-T (VANILLA) | 0.027 | 0.051 | 0.023 | 0.061 | 0.011 | 0.024 |

## A.6 FULL BAYESIAN TREATMENT WITH INFERENCE NETWORKS AND ABLATION STUDIES

Besides MAP estimates, we could also develop a full Bayesian treatment of ZESREC using variational inference networks as in typical variational autoencoders (VAE) Kingma & Welling (2014).

Specifically, using Jensen's inequality we have the following evidence lower bound (ELBO):

$$\log p(\mathbf{R}|\mathbf{X}, \lambda_u, \lambda_v) \geq E_q[\log p(\mathbf{R}, \mathbf{U}, \mathbf{V}|\mathbf{X}, \lambda_u, \lambda_v)] - E_q[\log q(\mathbf{U}, \mathbf{V}|\mathbf{X})], \qquad (3)$$

where the expectation is over $q(\mathbf{U}, \mathbf{V}|\mathbf{X})$. For the variational distribution $q(\mathbf{U}, \mathbf{V}|\mathbf{X})$ we have the following factorization:

$$q(\mathbf{U}, \mathbf{V}|\mathbf{X}) = q(\mathbf{V}|\mathbf{X})q(\mathbf{U}|\mathbf{V}, \mathbf{X}) = q(\mathbf{V}|\mathbf{X})q(\mathbf{U}|\mathbf{V}), \qquad (4)$$

where $\mathbf{U} = [\mathbf{u}_i]_{i=1}^I$, $\mathbf{V} = [\mathbf{v}_j]_{j=1}^J$, and $\mathbf{X} = [\mathbf{x}_j]_{j=1}^J$.

$$q(\mathbf{v}_j|\mathbf{X}) = q(\mathbf{v}_j|\mathbf{x}_j) = \mathcal{N}\Big(\mathbf{v}_j; f_{e,\mu}(\mathbf{x}_j), f_{e,\sigma^2}(\mathbf{x}_j)\Big),$$

$$q(\mathbf{u}_i|\mathbf{V}, \mathbf{X}) = \mathcal{N}\Big(\mathbf{u}_i; f_{seq,\mu}(\{\mathbf{v}_{j_t}\}_{t=1}^{n_i}), f_{seq,\sigma^2}(\{\mathbf{v}_{j_t}\}_{t=1}^{n_i})\Big). \qquad (5)$$

In other words, the inference network runs a similar Generative Process as the MAP approach, but replaces the prior $\mathbf{v}_j$ and $\mathbf{u}_i$ generators with $q(\mathbf{v}_j|\mathbf{X})$ and $q(\mathbf{u}_i|\mathbf{V}, \mathbf{X})$, respectively.

The training philosophy of VAE is different from MAP, but we will show that the end results look remarkably similar. To begin with, the first part of the ELBO wraps the original likelihood under the expectation over the variational distribution $q(\mathbf{U}, \mathbf{V}|\mathbf{X})$:

$$E_q[\log p(\mathbf{R}, \mathbf{U}, \mathbf{V}|\mathbf{X}, \lambda_u, \lambda_v)] = E_q[\log p(\mathbf{R}|\mathbf{U}, \mathbf{V}) + \log p(\mathbf{V}|\mathbf{X}) + \log p(\mathbf{U}|\mathbf{V})],$$

omitting dependencies when obvious. While the reconstruction likelihood for $p(\mathbf{R}|\mathbf{U}, \mathbf{V})$ stays the same, the other factors in the first part of the ELBO, $p(\mathbf{U}|\mathbf{X})$ and $p(\mathbf{V}|\mathbf{U})$, can be paired with the inference networks, $q(\mathbf{U}|\mathbf{X})$ and $q(\mathbf{V}|\mathbf{U})$, in the second part of the ELBO to produce a form of Kullback-Leibler (KL) divergence, e.g.,

$$-E_q[\log p(\mathbf{v}_j|\mathbf{x}_j) - \log q(\mathbf{v}_j|\mathbf{x}_j)] = D_{KL}(q(\mathbf{v}_j|\mathbf{x}_j)\|p(\mathbf{v}_j|\mathbf{x}_j))$$

$$= D_{KL}(\mathcal{N}(f_{e,\mu}(\mathbf{x}_j), f_{e,\sigma^2}(\mathbf{x}_j))\|\mathcal{N}(f_e(\mathbf{x}_j), \lambda_v^{-1}\mathbf{I}_D)), \qquad (6)$$

and similarly for the user-variable distributions. Since we choose Gaussian distributions, the KL-divergence yields explicit solution

$$(6) = \tfrac{1}{2}\Big[\lambda_v\|f_{e,\mu}(\mathbf{x}_j) - f_e(\mathbf{x}_j)\|_2^2 + \lambda_v\mathbf{1}^\top f_{e,\sigma^2}(\mathbf{x}_j) - \mathbf{1}^\top \log(\lambda_v f_{e,\sigma^2}(\mathbf{x}_j)) - D\Big].$$

Putting everything together, we may rewrite the negation of the ELBO (NELBO) as:

$$\mathcal{L}_e = E_q\left[-\sum_{i=1}^{I_s}\sum_{t=1}^{N_i}\log(f_{softmax}(\mathbf{u}_{it}^T\mathbf{v}_{i_t}))\right] + \text{Const.}$$

$$+ \tfrac{1}{2}E_{q(\mathbf{V}|\mathbf{X})}\sum_{i=1}^{I_s}\sum_{t=1}^{N_i}\Big[\lambda_u\|f_{seq,\mu}(\{\mathbf{v}_{i_\tau}\}_{\tau=1}^{t-1}) - f_{seq}(\{\mathbf{v}_{i_\tau}\}_{\tau=1}^{t-1})\|_2^2\Big]$$

$$+ \tfrac{1}{2}E_{q(\mathbf{V}|\mathbf{X})}\sum_{i=1}^{I_s}\sum_{t=1}^{N_i}\Big[\lambda_u\mathbf{1}^\top f_{seq,\sigma^2}(\{\mathbf{v}_{i_\tau}\}_{\tau=1}^{t-1}) - \mathbf{1}^\top \log(\lambda_u f_{seq,\sigma^2}(\{\mathbf{v}_{i_\tau}\}_{\tau=1}^{t-1}))\Big]$$

$$+ \tfrac{1}{2}\sum_j \Big[\lambda_v\|f_{e,\mu}(\mathbf{x}_j) - f_e(\mathbf{x}_j)\|_2^2 + \lambda_v\mathbf{1}^\top f_{e,\sigma^2}(\mathbf{x}_j) - \mathbf{1}^\top \log(\lambda_v f_{e,\sigma^2}(\mathbf{x}_j))\Big]$$

See Appendix A.10 for detailed derivation.

Additionally, if one assumes latent user vectors with zero variance, ignoring all related regularization terms, we may arrive at a simplified NELBO:

$$\mathcal{L}_e = E_q\left[-\sum_{i=1}^{I_s}\sum_{t=1}^{N_i}\log(f_{softmax}(\mathbf{u}_{it}^T\mathbf{v}_{i_t}))\right] + \tfrac{\lambda_v}{2}\|f_{e,\mu}(\mathbf{x}_j) - f_e(\mathbf{x}_j)\|_2^2$$

$$+ \tfrac{1}{2}\sum_j \Big[\lambda_v\mathbf{1}^\top f_{e,\sigma^2}(\mathbf{x}_j) - \mathbf{1}^\top \log(\lambda_v f_{e,\sigma^2}(\mathbf{x}_j))\Big] + \text{Const.}, \qquad (7)$$

where $f_e(\mathbf{x}_j)$ is the universal item embedding as part of the prior.

Table 3: Ablation studies on the full Bayesian treatment.

| Method | MIND N $\rightarrow$ F | | MIND L $\rightarrow$ F | |
|---|---|---|---|---|
| | N@20 | R@20 | N@20 | R@20 |
| ZESREC-H (BAYESIAN, MAP) | 0.029 | 0.079 | 0.025 | 0.075 |
| ZESREC-H (BAYESIAN, FULL) | **0.030** | **0.082** | **0.026** | **0.079** |

Finally, if we further ignore the variance for both the users and items, choosing deterministic $\mathbf{u}_{it} = f_{seq,\mu}(\{\mathbf{v}_{i_\tau}\}_{\tau=1}^{t-1})$ and $\mathbf{v}_j = f_{e,\mu}(\mathbf{x}_j)$, we may connect NELBO to the MAP objective, as promised.

**Ablation Studies on the Full Bayesian Treatment.** Table 3 shows the NDCG@20 and Recall@20 for our ZESREC-H variant under two different Bayesian inference schemes, MAP and the full Bayesian treatment. These results verify that our full Bayesian treatment for ZESREC can further improve zero-shot performance.

## A.7 IMPLEMENTATION DETAILS

We use pre-trained *google/bert_uncased_L-12_H-768_A-12* BERT model from Huggingface Wolf et al. (2020) to process item description and generate item embedding. The dimension of BERT embedding is 768. We use BERT embedding as input to a single-layer neural netwrok (NN) and the output dimension for the NN is set to $D$, which equals to the hidden dimension of the sequential model.

For ZESREC variants we use the default optimal setting: we set the hidden dimension $D$ as 300, the dropout rate as 0.2, and the number of training epochs as 20. We use Adagrad Défossez et al. (2020) as the optimizer with a learning rate of 0.1, and train ZESREC variants in the source domain with early stopping based on validation loss. We set the hyperparameter $\lambda_v$ as a relatively large value 100 to restrain the variance of the item offset vector $\boldsymbol{\epsilon}_j$.

For base models (HRNN, TCN, GRU4Rec) and corresponding base-meta models (HRNN-Meta, TCN-Meta, GRU4Rec-Meta), we set the dropout rate as 0.2 and the number of training epochs as 20; we choose Adagrad Défossez et al. (2020) as the optimizer. We train base models and base-meta models in the target domain with early stopping and perform hyperparameter tuning, both are based on the validation loss. We tried the hidden dimension $D$ in $\{128, 300\}$ and the learning rate $\eta$ in $\{0.01, 0.1, 1\}$, and choose to use the configurations $\{D : 128, \eta : 1\}$ for base models and $\{D : 128, \eta : 0.1\}$ for base-meta models.

For all datasets, we treat the rating as implicit feedback (interactions between user and item). Since we are considering session-based recommendation and using sequential model, we filter out users with only 1 interaction as the sequential model need to ingest at least one item from user history as context to perform next-step prediction.

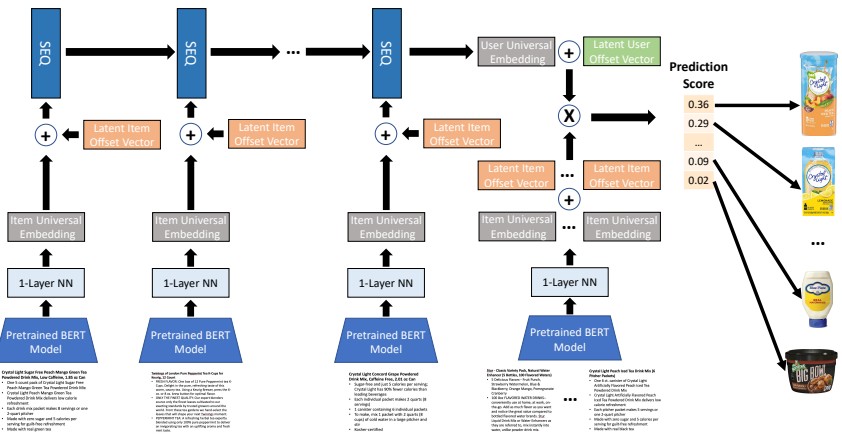

Figure 5: Model architecture for the simplified ZESREC (without Bayesian inference).

All experiments were ran on a GPU machine with Nvidia Tesla V100 16G memory GPU.

## A.8 MODEL ARCHITECTURE

The graphical model for ZESREC is shown in figure 1. In the MAP estimation version of ZES-REC, we set $\lambda_u \to \infty$ to remove latent user offset vector, thereby preventing ZESREC from over-parameterization. figure 5 shows a simplified deterministic model architecture from the neural network point of view. Below we elaborate on the process in terms of two stages: training in source domain and inference in target domain.

**Training in Source Domain.** During training, on the encoder side, we generate BERT embeddings from items' NL descriptions as item universal embeddings and then add the learnable item offset vectors to them, which yield the final item embeddings (item latent vectors). The sequential model will aggregate item embeddings of items in user history to generate user embeddings. On the decoder side, we compute item latent vectors the same way we do on the encoder side. Here we share item latent vectors on both encoder and decoder sides to reduce number of parameters. Empirically we find this prevent overfitting and improve performance.

**Inference in Target Domain.** In this phase, we will remove the item offset vectors on both the encoder and decoder sides. We use item universal embeddings directly instead as the final item latent vectors since we have no access to interactions in the target domain, and therefore the model is unable to estimate the item offset vectors.

**Item Universal Embedding Network (Item UEN).** Item universal embedding network $f_e(\cdot)$ extracts item $j$'s embedding $\mathbf{m}_j$ based on its NL description $x_j$, while $\mathbf{m}_j$ is generalizable across domains. The network consists of a pretrained BERT network $f_{BERT}$ followed by a single-layer neural network $f_{NN}(\cdot)$ which is used to adapt the pre-trained BERT for recommendation tasks: $\mathbf{m}_j = f_e(\mathbf{x}_j) = f_{NN}(f_{BERT}(\mathbf{x}_j))$. Note that we use the 'CLS' token embedding as the output of $f_{BERT}(.)$ and item UEN is jointly trained with the sequential model using the objective function in Eqn. 1.

**User Universal Embedding Network (User UEN).** The user UEN $f_{seq}(\cdot)$ is built on top of the item UEN above which is an aggregation function (RNN, CNN, etc.) over universal item embeddings in user history. Note that this user UEN is used during both training and inference.

## A.9 DATA USAGE AND PRIVACY DISCUSSIONS

**Data Usage.** For real-world scenarios, new business owners who hope to do zero-shot recommendation need to check the dataset policy before usage. On the other hand, if multinational corporations hope to establish a branch in a new region they could use their own data from other existing regions.

**Privacy.** To protect the privacy of user data, we encourage people who want to adopt our methods to train the model using source-domain data with differential privacy embedded. Some related references include (Abadi et al., 2016; Chen et al., 2019; Wang et al., 2019).

## A.10 MORE DETAILED DERIVATION FOR THE FULL BAYESIAN TREATMENT WITH INFERENCE NETWORKS

**Training.** Using Jensen's inequality we have the following evidence lower bound (ELBO):

$$\log p(\mathbf{R}|\mathbf{X}, \lambda_u, \lambda_v) \geq E_q[\log p(\mathbf{R}, \mathbf{U}, \mathbf{V}|\mathbf{X}, \lambda_u, \lambda_v)] - E_q[\log q(\mathbf{U}, \mathbf{V}|\mathbf{X})],$$

where the expectation is over $q(\mathbf{U}, \mathbf{V}|\mathbf{X})$. For the variational distribution $q(\mathbf{U}, \mathbf{V}|\mathbf{X})$ we have the following factorization:

$$q(\mathbf{U}, \mathbf{V}|\mathbf{X}) = q(\mathbf{V}|\mathbf{X})q(\mathbf{U}|\mathbf{V}, \mathbf{X}) = q(\mathbf{V}|\mathbf{X})q(\mathbf{U}|\mathbf{V}), \tag{8}$$

where $\mathbf{U} = [\mathbf{u}_i]_{i=1}^I$, $\mathbf{V} = [\mathbf{v}_j]_{j=1}^J$, and $\mathbf{X} = [\mathbf{x}_j]_{j=1}^J$

$$q(\mathbf{v}_j|\mathbf{X}) = q(\mathbf{v}_j|\mathbf{x}_j) = \mathcal{N}\Big(\mathbf{v}_j; f_{e,\mu}(\mathbf{x}_j), f_{e,\sigma^2}(\mathbf{x}_j)\Big),$$

$$q(\mathbf{u}_i|\mathbf{V}, \mathbf{X}) = \mathcal{N}\Big(\mathbf{u}_i; f_{seq,\mu}(\{\mathbf{v}_{j_t}\}_{t=1}^{n_i}), f_{seq,\sigma^2}(\{\mathbf{v}_{j_t}\}_{t=1}^{n_i})\Big). \tag{9}$$

With these variational distributions, we can then expand the ELBO, denoted as $\mathcal{L}_e$, as:

$$\mathcal{L}_e = E_q[\log p(\mathbf{R}, \mathbf{U}, \mathbf{V}|\mathbf{X}, \lambda_u, \lambda_v)] - E_q[\log q(\mathbf{U}, \mathbf{V}|\mathbf{X})] \tag{10}$$

$$= E_q[\sum_{i=1}^{I_s} \sum_{t=1}^{N_i} \log(f_{softmax}(\mathbf{u}_{it}^T \mathbf{v}_{i_t}))]$$

$$+ E_q[\log p(\mathbf{V}|\mathbf{X})] + E_q[\log p(\mathbf{U}|\mathbf{V})] - E_q[\log q(\mathbf{U}, \mathbf{V}|\mathbf{X})] + C,$$

where the expectation is over $q(\mathbf{U}, \mathbf{V}|\mathbf{X})$ in Eqn. 8, and $C$ is a constant.

Below we discuss how to compute the three terms $E_q[\log p(\mathbf{V}|\mathbf{X})]$, $-E_q[\log q(\mathbf{U}, \mathbf{V}|\mathbf{X})]$ and $E_{q(\mathbf{U}, \mathbf{V}|\mathbf{X})}[\log p(\mathbf{U}|\mathbf{V})]$ in detail.

**Computing $E_q[\log p(\mathbf{V}|\mathbf{X})]$.** We can compute $E_q[\log p(\mathbf{V}|\mathbf{X})]$ in closed form as (we omit the constant $\log \sqrt{2\pi}$ for clarity):

$$E_q[\log p(\mathbf{V}|\mathbf{X})] = \tfrac{1}{2} D \log \lambda_v - \tfrac{\lambda_v}{2}[\sum_j \|f_{e,\sigma^2}(\mathbf{x}_j)\|_1 + \|f_{e,\mu}(\mathbf{x}_j) - \mathbf{m}_j\|_2^2],$$

where $\mathbf{m}_j$ is the item university embedding and also the mean of $p(\mathbf{v}_j|\mathbf{X})$, as defined in section 2.2.
**Computing $E_q[\log q(\mathbf{U}, \mathbf{V}|\mathbf{X})]$.** The term $-E_q[\log q(\mathbf{U}, \mathbf{V}|\mathbf{X})]$ is the entropy of $q(\mathbf{U}, \mathbf{V}|\mathbf{X})$, denoted as $H[q(\mathbf{U}, \mathbf{V}|\mathbf{X})]$ below. We have

$$H[q(\mathbf{U}, \mathbf{V}|\mathbf{X})] = -E_{q(\mathbf{V}|\mathbf{X})} E_{q(\mathbf{U}|\mathbf{V})}[\log q(\mathbf{V}|\mathbf{X}) + \log q(\mathbf{U}|\mathbf{V})]$$

$$= -E_{q(\mathbf{V}|\mathbf{X})} E_{q(\mathbf{U}|\mathbf{V})}[\log q(\mathbf{V}|\mathbf{X})] - E_{q(\mathbf{V}|\mathbf{X})} E_{q(\mathbf{U}|\mathbf{V})}[\log q(\mathbf{U}|\mathbf{V})]$$

$$= -E_{q(\mathbf{V}|\mathbf{X})}[\log q(\mathbf{V}|\mathbf{X})] - E_{q(\mathbf{V}|\mathbf{X})}\Big[E_{q(\mathbf{U}|\mathbf{V})}[\log q(\mathbf{U}|\mathbf{V})]\Big]$$

$$= \tfrac{1}{2} \sum_j [\mathbf{1}^\top \log f_{e,\sigma^2}(\mathbf{x}_j)] + \tfrac{1}{2} \sum_i E_{q(\mathbf{V}|\mathbf{X})}\Big[\sum_{t=1}^{N_i} \mathbf{1}^\top \log f_{seq,\sigma^2}(\{\mathbf{v}_{i_\tau}\}_{\tau=1}^{t-1})\Big] + C$$

$$\approx \tfrac{1}{2} \sum_j [\mathbf{1}^\top \log f_{e,\sigma^2}(\mathbf{x}_j)] + \tfrac{1}{2N_v} \sum_i \sum_{\mathbf{V}} \Big[\sum_{t=1}^{N_i} \mathbf{1}^\top \log f_{seq,\sigma^2}(\{\mathbf{v}_{i_\tau}\}_{\tau=1}^{t-1})\Big] + C, \tag{11}$$

where in the last line $\mathbf{V}$ is sampled for $N_v$ times to get a Monte Carlo estimate of $E_{q(\mathbf{V}|\mathbf{X})}[\log f_{seq,\sigma^2}(\mathbf{v}_{i_\tau}\}_{\tau=1}^{t-1})]$. In practice, it is found that one sample is usually sufficient due to the use of SGD-based optimization process Kingma & Welling (2014).

**Computing $E_{q(\mathbf{U}, \mathbf{V}|\mathbf{X})}[\log p(\mathbf{U}|\mathbf{V})]$.** Similar to the processing of computing $E_q[\log p(\mathbf{V}|\mathbf{X})]$ above and omitting the constants $\log \sqrt{2\pi}$ and $D \log \lambda_u$ for clarity, we have

$$E_q[\log p(\mathbf{U}|\mathbf{V})]$$

$$= E_{q(\mathbf{V}|\mathbf{X})}\Big[E_{q(\mathbf{U}|\mathbf{V})}[\log p(\mathbf{U}|\mathbf{V})]\Big]$$

$$= -\tfrac{\lambda_u}{2} E_{q(\mathbf{V}|\mathbf{X})}\Big[\sum_{i=1}^{I_s} \sum_{t=1}^{N_i} [\|f_{seq,\sigma^2}(\{\mathbf{v}_{i_\tau}\}_{\tau=1}^{t-1})\|_1 + \|f_{seq,\mu}(\{\mathbf{v}_{i_\tau}\}_{\tau=1}^{t-1}) - \mathbf{n}_{it}\|_2^2]\Big]$$

$$\approx -\tfrac{\lambda_u}{2N_v} \sum_{\mathbf{V}} \sum_{i=1}^{I_s} \sum_{t=1}^{N_i} [\|f_{seq,\sigma^2}(\{\mathbf{v}_{i_\tau}\}_{\tau=1}^{t-1})\|_1 + \|f_{seq,\mu}(\{\mathbf{v}_{i_\tau}\}_{\tau=1}^{t-1}) - \mathbf{n}_{it}\|_2^2], \tag{12}$$

where $\mathbf{n}_{it}$ is the user $i$'s universal embedding at time $t$ and also the mean of user latent vector distribution $p(\mathbf{u}_{it}|\{\mathbf{v}_{i_\tau}\}_{\tau=1}^{t-1})$, as defined in section 2.2. Similar to Eqn. 11, $\mathbf{V}$ is sampled for $N_v$ times from $q(\mathbf{V}|\mathbf{X})$ to get a Monte Carlo estimate of $E_{q(\mathbf{V}|\mathbf{X})}[\|f_{seq,\sigma^2}(\mathbf{v}_{i_k}\}_{k=1}^{n_i})\|_2^2]$.

**Latent User Vectors with Zero Variance.** If one assumes latent user vectors with zero variance, i.e., $\lambda_u = \infty$, the variational distribution for $u_i$ can be set to zero variance as well. Specifically, Eqn. 9 becomes

$$q(\mathbf{u}_i|\mathbf{V}, \mathbf{X}) = \mathcal{N}\Big(\mathbf{u}_i; f_{seq,\mu}(\{\mathbf{v}_{jt}\}_{t=1}^{n_i}), \mathbf{0}\Big).$$

Correspondingly, the term $E_q[\log q(\mathbf{U}, \mathbf{V}|\mathbf{X})]$ in Eqn. 11 becomes

$$E_q[\log q(\mathbf{U}, \mathbf{V}|\mathbf{X})] = \tfrac{1}{2} \sum_j [\mathbf{1}^\top \log f_{e,\sigma^2}(\mathbf{x}_j)],$$

Similarly, in Eqn. 12 the term $E_{q(\mathbf{U},\mathbf{V}|\mathbf{X})}[\log p(\mathbf{U}|\mathbf{V})]$ becomes 0, and $f_{seq,\mu}(\{\mathbf{v}_{i_\tau}\}_{\tau=1}^{t-1}) = \mathbf{n}_{it}$. The ELBO in Eqn. 10 then becomes

$$\mathcal{L}_e = E_q[\sum_{i=1}^{I_s}\sum_{t=1}^{N_i}\log(f_{softmax}(\mathbf{u}_{it}^T\mathbf{v}_{i_t}))] - \frac{\lambda_v}{2}[\sum_j \|f_{e,\sigma^2}(\mathbf{x}_j)\|_1 + \|f_{e,\mu}(\mathbf{x}_j) - \mathbf{m}_j\|_2^2] + \frac{1}{2}\sum_j[\mathbf{1}^\top\log f_{e,\sigma^2}(\mathbf{x}_j)]$$

**Inference.** Inference can be done via Monte Carlo estimates of $p(\mathbf{R}|\mathbf{X}, \lambda_u, \lambda_v)$. Specifically,

$$p(\mathbf{R}|\mathbf{X}, \lambda_u, \lambda_v) = E_q(p(\mathbf{R}|\mathbf{U},\mathbf{V})) \approx \frac{1}{N_v N_u}\sum_{\mathbf{V}^{(n)}}\sum_{\mathbf{U}^{(n)}} p(\mathbf{R}|\mathbf{U}^{(n)}, \mathbf{V}^{(n)}),$$

where $\mathbf{V}^{(n)} \sim q(\mathbf{V}|\mathbf{X})$ and $\mathbf{U}^{(n)} \sim q(\mathbf{U}|\mathbf{V}^{(n)}, \mathbf{X})$.

One could also use MAP inference to trade accuracy for speed.

$$p(\mathbf{R}|\mathbf{X}) = \int p(\mathbf{R}|\mathbf{U},\mathbf{V},\mathbf{X})p(\mathbf{U},\mathbf{V}|\mathbf{X})d\mathbf{U}d\mathbf{V}$$

$$\approx \int p(\mathbf{R}|\mathbf{U},\mathbf{V},\mathbf{X})\delta_{\mathbf{U}_{MAP}}(\mathbf{U})\delta_{\mathbf{V}_{MAP}}(\mathbf{V})d\mathbf{U}d\mathbf{V},$$

where $\delta(\cdot)$ denotes a Dirac delta distribution. $\mathbf{U}_{\text{MAP}}$ and $\mathbf{V}_{\text{MAP}}$ are the MAP estimate of $\mathbf{U}$ and $\mathbf{V}$ given $\mathbf{X}$:

$$(\mathbf{U}_{MAP}, \mathbf{V}_{MAP}) \approx \operatorname*{argmax}_{\mathbf{U},\mathbf{V}} q(\mathbf{U},\mathbf{V}|\mathbf{X}) = \Big(f_{seq}(f_{e,\mu}(\mathbf{X})), f_{e,\mu}(\mathbf{X})\Big).$$

