# OpenReview forum: "Zero-Shot Recommender Systems"
_ICLR.cc/2022/Workshop/DGM4HSD — ICLR 2022 DGM4HSD workshop Poster_

### Official Review · Reviewer_6hEy · 2022-03-20

**Rating:** 7
**Confidence:** 3

**Review:**

**Summary**: The paper proposes ZESREC, a probabilistic framework that enables zero-shot learning for recommender systems. The method compares favorably to existing recommender system approaches in zero-shot learning scenarios without user nor item overlaps.

**Strengths**
- The proposed universal item embeddings do not depend on user-item interactions, which allows transferring knowledge across domains.
- The Bayesian approach appears to be flexible and well-grounded.
- Proposed method achieves performance comparable to in-domain baselines (i.e. baselines trained directly on the target dataset), highlighting the generality and transferability of the approach.

**Weaknesses**
- No error bars in Table 1.

The approach seems original and of relevance to the workshop. Therefore I recommend the acceptance of this paper.

---

### Official Review · Reviewer_GRME · 2022-03-27
**Solid contribution**

**Rating:** 8
**Confidence:** 4

**Review:**

This paper studies cross-domain recommendation with zero-shot learning settings. Specifically, this work studies cold item/user, domain gaps, and no accessible target data settings. A generative model leveraging natural language description as countinuous item ID is proposed. Experiments on real-world datasets demonstrate the effectiveness.

Strengths:
+ The work studies a fully cold start scenario for sequential recommendation. I believe the problem setup is novel and has praticle values to real-world deployment of recommender systems.
+ The writing is clear and the idea is easy to follow.

Weaknesses:
- The authors may elaborate how to infer user embeddings in the target domain.
- Real-world A/B tests could be conducted if possible.

---

### Decision · Program_Chairs · 2022-03-27

Accept (Poster)